

# The sensitivity of the large-scale atmosphere circulation to changes in surface temperature gradients in the Northern Hemisphere

Sonja Molnos[1,2], Stefan Petri[1], Jascha Lehmann[1], Erik Peukert[1,2], Dim Coumou[1,3]

[1] Potsdam Institute for Climate Impact Research, Potsdam, Germany
[2] Department of Physics, University of Potsdam, Germany
[3] Institute for Environmental Studies (IVM), VU University Amsterdam

*Correspondence to*: S. Molnos (molnos@pik-potsdam.de)

**Abstract.**

Climate and weather conditions in the mid-latitudes are strongly driven by the large-scale atmosphere circulation. Observational data indicates that important components of the large-scale circulation have changed in recent decades including the strength of the Hadley cell, jet streams, storm tracks and planetary waves. Associated impacts cover a broad range, including changes in the frequency and nature of weather extremes and shifts of fertile habitats with implications for biodiversity and agriculture. Dynamical theories have been proposed that link the shift of the poleward edge of the Northern Hadley cell to changes in the meridional temperature gradient. Moreover, model simulations have been carried out to analyse the cause of observed and projected changes in the large-scale atmosphere circulation. However, the question of the underlying drivers and particularly the possible role of global warming is still debated. Here, we use a statistical-dynamical atmosphere model (SDAM) to analyse the sensitivity of the Northern Hemisphere Hadley cell, storm tracks, jet streams and planetary waves to changes in temperature fields by systematically altering the zonal and meridional temperature gradient as well as the global mean surface temperature.

SDAMs are computationally fast compared to more complex general circulation models (GCM) which enables us to scan a large and high-dimensional parameter space for sensitivity analyses using more than thousand individual model runs.

Our results show that the strength of the Hadley cell, storm tracks and jet streams depends almost linearly on both the global mean temperature and the meridional temperature gradient whereas the zonal temperature gradient has little or no influence. The magnitude of planetary waves is clearly affected by all three temperature components. Finally, the width of the Hadley cell behaves nonlinearly with respect to all three temperature components.

Under global warming the meridional temperature gradient is expected to change: Enhanced warming is expected in the Arctic, largely near the surface, and at the equator at high altitudes. Also there is a pronounced seasonality to these warming patterns. Using SDAMs to disentangle and separately analyse the effect of individual temperature changes might thus help to understand observed and projected changes in large-scale atmosphere dynamics.



**Keywords:**

Earth System Model of Intermediate Complexity, sensitivity experiments, Statistical-dynamical atmosphere models, Hadley cell, Jet stream, circulation changes

## 1    Introduction

Large-scale atmosphere dynamics including Hadley cells, jet streams, storm tracks and planetary waves play a key role in the general circulation of the atmosphere, determining climatic conditions worldwide.

Hadley cells are large-scale tropical atmospheric circulations and responsible for the transport of heat and moisture from the equator to the mid-latitudes (D'Agostino and Lionello, 2016). Recent literature suggests that the Hadley cells have widened

in the past few decades due to ozone depletion in the Southern Hemisphere (Kang et al., 2011; Son et al., 2009) and due to increases in heterogeneous warming agents as carbon dioxide and tropospheric ozone in the Northern Hemisphere (Allen et al., 2012). Further possible drivers of the Hadley cell are sea surface temperature variations, which may lead to tropical contraction (Allen et al., 2014), stratospheric cooling, global warming or changes in baroclinic eddy phase speeds (Chen et al., 2014). While most studies based on reanalysis products observe that together with this widening there is also a

strengthening of the Hadley cells (Nguyen et al., 2013), models project a weakening over the 21$^{st}$ century under a high emission scenario (Lu et al., 2007). Those discrepancies may be explained by relatively short observational records, large natural variability or model deficiencies (Allen et al., 2014).

The strength and the width of the Hadley cell circulation have strong implications for a variety of atmospheric phenomena such as jet streams, extratropical storms and planetary waves.

Jet streams are upper-level fast currents of westerly winds that circulate around the hemisphere. Under climate change a weakening of the northern subtropical jet stream in winter is observed by most studies (Molnos et al., 2017; Rikus, 2015) though a few studies observe also a positive trend (Pena-Ortiz et al., 2013). The underlying drivers for those changes are still debated. One possible driver is the Arctic amplification leading to a decreasing meridional temperature gradient which might be responsible for a poleward movement of the jet stream.

Storm tracks play a crucial role in modulating precipitation in the Earth system. Yin et al. (2005) studied storm tracks under climate change using 15 coupled climate models and found that storm tracks shift poleward and intensify under climate change (Yin, 2005). In addition, O'Gorman (2010) discovered that the storm track intensity is nonlinearly related to global warming. In contrast to these changes in the large-scale circulation components, a recent study by Barnes (2013) indicates that there are no significant trends in the strength of long-term quasi-stationary planetary waves under climate change. These

waves strongly interact with storm track activity in the mid-latitudes. Coumou et al. (2015) show that during the satellite period (1979-2013) atmospheric circulation has significantly weakened in boreal summer. They argue that the weakening





might be related to a decreasing meridional temperature gradient. Consistently, CMIP5 models project a further decrease in summer storm track activity under a high-emission scenario until the end of the 21$^{st}$ century (Lehmann et al., 2014).

All the above mentioned studies have in common that for the analyses both zonal and meridional temperature gradients as well as global mean temperature are changed simultaneously over time. For that reason changes in Hadley cell, jet streams, planetary waves and storm tracks cannot be directly assigned to individual temperature components.

For the analysis presented here we use the statistical-dynamical atmosphere model Aeolus 1.0 (Molnos et al., 2016). It is a 2.5 – dimensional model with the vertical dimension coarsely resolved (5 dynamical layers in the troposphere) and therefore belongs to the model class of intermediate complexity atmosphere models (Claussen et al., 2002; Petoukhov et al., 2000). Aeolus is based on time-averaged equations in which transient eddies are parameterized in terms of the large scale field (Coumou et al., 2011; Peixoto and Oort, 1992; Saltzman, 1978). This means that instead of solving every eddy directly, only the ensemble mean eddy characteristics (in terms of heat, water vapour and momentum transport) are solved. The essential difference compared to more widely used general circulation models (GCMs) is thus the point of truncation in the frequency spectrum of atmospheric motion (Saltzman, 1978). This different approach allows much coarser spatial and temporal discretization, making SDAMs computationally efficient. Here we use the model to separately analyse the influence of changes in global-mean surface temperature and zonal and meridional temperature gradients on large-scale atmosphere dynamics in the Northern Hemisphere winter season.

## 2    Data and Methods

### 2.1    Aeolus forcing parameters

The simulations were forced by climatological (1979-2014) winter mean (December-January-February (DJF)) data of sea surface temperature and specific humidity, using ERA-Interim data from the European Centre for Medium-Range Weather Forecasts (ECMWF) (Dee et al., 2011). First, the data was regridded to 3.75° × 3.75° (longitude × latitude). Since large temperature gradients lead to numerical instabilities, we map 500 mb temperatures onto the sea level using a standard lapse rate equation to obtain smooth temperature profiles, as in Molnos et al. (2016).

The parameter values were taken from the calibration process, described in Molnos et al (2016), which optimizes the model's representation of the tropical large-scale circulation.

### 2.2    Temperature components



In this and the following chapters, the angle brackets denote time-averaged quantities, the overbar denotes zonal mean quantities, the prime indicates synoptic scale components (2 – 6 days period), and the star indicates azonal components, i.e. deviations from the zonal mean.

For the sensitivity analysis we vary three different temperature components: (1) the meridional temperature gradient $\frac{dT}{d\phi}$, (2) the azonal temperature $T^*$ (i.e. deviations from the zonal mean) and (3) the global mean temperature $T_{global}$.

We change the temperature for each grid cell with respect to parameters for the three components in two steps. First, the parameter $w_{dTd\phi}$ is used to vary the meridional temperature gradient by cooling/warming the poles and imposing the global mean temperature

$$T_1(\phi,\lambda) = T_{EQ}(\lambda) + \left(T_{DJF}(\phi,\lambda) - T_{EQ}(\lambda)\right) * w_{dTd\phi} + T_{global}, \tag{1}$$

whereby $\phi$ and $\lambda$ are respectively latitude and longitude, $T_{DJF}(\phi,\lambda)$ is the original temperature, $T_{EQ}(\lambda)$ the temperature at the equator and $T_1(\phi,\lambda)$ is the updated temperature.

In the second step, the parameter $w_{azonal}$ is used to alter the azonal temperature, which is added to the zonal mean temperature $\overline{T_1}$.

$$T_{New}(\phi,\lambda) = T_1^*(\phi,\lambda) * w_{azonal} + \overline{T_1}(\phi) \tag{2}$$

This way $w_{dTd\phi} = 1$, $w_{azonal} = 1$ and $T_{global} = 0$ indicate present day conditions. $T_{New}$ is then used as model input. The parameters $w_{dTd\phi}$ and $w_{azonal}$ are varied between 0.75–1.1 (with steps of 0.025) to examine the behavior of the dynamical core under conditions between $-25\%$ to $+10\%$ of their present-day wintertime climatological values. These limits roughly correspond to expected temperature gradients in a $2 \times CO_2$ scenario and during the last glacial maximum scenarios (Coumou et al., 2011). Large azonal temperature differences, i.e. large values of $w_{azonal}$, imply strong temperature deviations between land masses and oceans. Small $\frac{dT}{d\phi}$ values represent amplified warming of the poles, compared to the equator, and thus a reduced meridional temperature gradient.

The parameter $T_{global}$ is altered between $\pm 4K$ (with steps of 4K) of the present-day (PD) climatology (1979 – 2014) temperature. This range covers climate variability over the past million years and possible near future changes.

For each temperature component we determine its influence on the strength and width of the Hadley cell, as well as the strength of zonal-mean jets, storm tracks and planetary waves in the Northern Hemisphere.

To force the Aeolus dynamical core, we use perturbed surface temperature profiles derived from the ERA-Interim winter climatology as explained above. We perform 2601 simulations with a regular 3-dimensional parameter space using the multi-run simulation environment *SimEnv*, which provides a tool to inspect the model's behaviour in the parameter space by discrete numerical sampling (Flechsig et al., 2013).





### 2.3    Dynamical variables

To obtain the strength of the jet stream for this analysis, we use seasonally (DJF) averaged zonal mean zonal wind $\overline{\langle u \rangle}$. For simplicity, we define the jet stream position as the latitude of the maximum of $\overline{\langle u \rangle}$ between 10°N and 80°N at 9000 m height

(corresponding to a pressure level of ca. 300 mbar).

We define the width and the strength of the Hadley cell by calculating the integrated southward mass flux in the lower troposphere between 1000 mb and 500 mb from the zonal mean meridional wind velocity (Molnos et al, 2016).

As a measure of storm track activity we calculate the eddy kinetic energy $(E_K' = 0.5(u'^2 + v'^2))$ , whereby $u'^2$ and $v'^2$ are the zonal and meridional synoptic wind velocity, and use its maximum between 10° N and 80° N at 9000 m (ca. 300 mb) to

analyse the strength and shift of the storm track activity.

To analyse the planetary waves we calculate the average of all positive values between 20°N-80°N of the azonal wind components $\langle u^* \rangle$ and $\langle v^* \rangle$.

## 3    Results

We compare and analyze the zonal mean dynamical variables of eddy kinetic energy $\overline{\langle E_K' \rangle}$ (which captures storm track

activity), zonal mean zonal wind velocity $\overline{\langle u \rangle}$ and the vertical integral of the lower tropospheric integrated mass flux $\overline{\langle m \rangle}$ as well as azonal wind velocities $\langle u^* \rangle$ and $\langle v^* \rangle$.

### 3.1    Tropical circulation

#### 3.1.1    Strength and shift of the Hadley cell


The integrated mass flux in the lower troposphere of the present-day modelled climatological NH winter values (orange line in Fig. 1) captures well the shape of the red curve from ERA-Interim data. In particular, the maximum strength, defined as the minimum between the zero-crossings, is close to the ERA-Interim data. The modelled Hadley cell's width, defined as the distance between the mass flux zero-crossings near 0° and 30° latitude of the orange line, is smaller than in ERA-Interim

(red curve).

For further analysis we plot the width (Fig. 2) and strength (Fig. 3) of the Hadley cell as a function of $w_{dTd\phi}$ and $w_{azonal}$ for three different global mean temperature values ((a) $T_{global} = \mathrm{PD} - 4K$ , (b) $T_{global} = \mathrm{PD}$ and (c) $T_{global} = \mathrm{PD} + 4K$). In general, both a stronger meridional temperature gradient and a stronger azonal temperature contrast lead to a nonlinear



broadening of the Hadley cell width. However, for present day $T_{global}$ and $T_{global} = \text{PD}+ 4\text{K}$ there are critical values in $w_{dTd\phi} > 1.05$ and $w_{azonal} > 0.9$, where the width decreases slightly.

The Hadley cell width shows larger changes in response to changes in the meridional temperature gradient than for changes in the azonal temperature gradient, indicating that the former has a stronger relative influence.

The Hadley cell strengthens with increasing meridional gradient and only weakly depends on $w_{azonal}$ (Fig. 3), and depends stronger on global mean temperature $T_{global}$.

## 3.2   Extratropical circulation

### 3.2.1   Strength of the jet stream

The jet stream location and strength is visible as two distinct maxima in the zonal-mean zonal wind velocity in ERA-Interim (Fig. 4a). Aeolus reasonably reproduces the main jet stream features in terms of spatial position and magnitude (Fig. 4c). The modelled magnitude of the jet in the Northern Hemisphere is in better agreement with reanalysis data than in the Southern Hemisphere. This is likely related to the fact that Aeolus is not coupled to an ice model and thus effects from the Antarctic

ice sheet are not considered. The model reasonably reproduces near-surface tropical easterlies ("trade winds") at low latitude.

Fig. 4(b) and Fig. 4(d) show the impact of changes in the meridional temperature gradient $\frac{dT}{d\phi}$ on jet stream dynamics. With a higher meridional temperature gradients, the strength of the jet stream increases and with lower temperature gradient the strength decreases.

This is also observed in Figure 5 where the jet stream strength is shown as function of $w_{dTd\phi}$ and $w_{azonal}$ for three different

values of $T_{Global}$.

The strength of the jet stream is sensitive to the meridional temperature gradient ($w_{dTd\phi}$) and to the global mean temperature. The azonal temperature contrasts have little influence on the jet stream strength.

### 3.2.2   Strength of the storm track activity


The NH winter climatological (1979 - 2014) storm track's eddy kinetic energy in Aeolus (Fig. 6(c)) is similar to ERA-Interim data (Fig. 6(a)).



Figures 6(b)-(d) show that storm activity increases with increasing temperature gradient ((b) $w_{dTd\phi} = 0.75$, (c) $w_{dTd\phi} = 1$ and (d) $w_{dTd\phi} = 1.1$).

The strength of the storm track activity depends on both $w_{dTd\phi}$ and $w_{azonal}$ (Fig. 7) in a way that the influence of $w_{dTd\phi}$ dominates the influence on storm track activity. The global mean temperature leads to a general strengthening of the storm
track activity (Fig. 7).

### 3.2.3    Strength of the planetary waves

The strength of the planetary waves is roughly as sensitive to $w_{dTd\phi}$ as to $w_{azonal}$, both in terms of $\langle u^* \rangle$ [Fig. 8(a) − (c)] and in terms of $\langle v^* \rangle$ [Fig. 8(d) − (f)]. Both meridional and zonal wind directions exhibit the same relationship such that larger
meridional and azonal temperature gradients lead also to stronger winds. In addition, if the meridional temperature gradient is smaller than the azonal temperature gradient, the strength of the planetary waves increases faster with higher meridional temperature gradient then if both gradients have a similar magnitude and vice versa leading to a wave-shaped structure ([Fig. 8(a),(b),(c)] and [Fig. 8(d),(e),(f)]).

The global mean temperature has a positive but only weak influence on the strength of planetary waves.

## 15   4    Discussion

For all investigated atmosphere variables we observe a strengthening for higher global mean temperature and higher absolute meridional temperature gradients and only a weak (strong) dependence on the azonal temperature for storm tracks (planetary waves), which we discuss in comparison with results from literature in the following sections. However, most previous studies have analysed only the combined effect of changes in several temperature components making a direct comparison
difficult.

### 4.1    Tropical circulation

### 4.1.1    Strength and shift of the Hadley cell

Our analysis indicates that a higher absolute meridional and azonal temperature gradient leads to a larger Hadley cell width, and we observe only a very weak dependence on the global mean temperature.

Frierson et al. (2007) tested the sensitivity of the Hadley circulation to changes in temperature gradient and global mean temperature only. Consistent with our results, they found that temperature gradients in both an idealized moist GCM and a full GCM lead to a widening of the Hadley cell. They concluded that the reason for this behaviour is the increased static



stability with increasing global mean temperature and meridional temperature gradient. In addition, they found that the Hadley cell width does not depend strongly on model physics, because their simulation results are similar, independent of the physical parameterizations.

Adam et al. (2014) examined the Hadley cell using 6 reanalysis datasets, 22 Atmospheric Modeling Intercomparison Project

(AMIP) simulations, and 11 historical Ocean-Atmosphere coupled simulations from phase 5 of the Climate Modeling Intercomparison Project (CMIP5). To distinguish between temperature gradient and global mean temperature, they decomposed sea surface temperature (SST) into factors that are primarily associated with global warming (mean SST changes) and ENSO (SST gradient changes). They concluded that a weakening of the temperature gradient and a strengthening of the global mean temperature leads to a widening of the Hadley cell. This thus differs from the findings of

Frierson et al. (2007) and ours.

Lu et al. (2007) analysed both subtropical dry zone and widening of the Hadley cell in the climate change simulations of the IPCC AR4 project. Hereby, increased GHGs were used to change the temperature. According to their results, a higher global mean temperature leads to a weakening and widening of the Hadley cell.

Again, it should be emphasized these other studies did not strictly isolate between different causes (meridional temperature

gradient, the global mean temperature or the azonal temperature contrast) that can affect the Hadley circulation. Most likely Lu et al. (2007) also changed the meridional temperature gradient with increasing global mean temperature. With decreasing temperature gradient and increasing global mean temperature, we also find a weakening Hadley cell. In addition, Lu et al. (2007) observed a widening of the Hadley cell. In our study, a rising global mean temperature and a strengthening of the azonal temperature component would lead to a widening of the Hadley cell.

In order to quantify the influence of the temperature gradient, we examine the poleward edge of the Hadley cell, defined as the zero crossing point of the mass flux close to 30°N, depending on $w_{dTd\phi}$. We compare our results with the results of the theoretical poleward edge of the Hadley cell from Held and Hou. They developed a theory for an axially symmetric circulation and stratified rotating Boussinesq fluid and derived, among others, an equation for the poleward edge of the Hadley cell (Held and Hou, 1980). Our and their results are qualitatively similar and exhibit the same trend (Peukert, 2015).


In agreement to our results, Mitas and Clement (2005) detected a strengthening of the Hadley cell in their analyses even though they found great differences between different data sets. They used several reanalysis data sets as well as a rawsonde data set and a model data set.

## 4.2    Extratropical circulation

### 4.2.1    Strength of the jet stream



We show that the strength of the jet stream decreases with decreasing absolute meridional temperature gradient (Fig. 4 and Fig. 5).

This is in agreement with findings from Polvani and Kushner (2002) and Haigh et al (2005.) Polvani and Kushner (2002) used a simple general circulation model and showed that for sufficiently strong cooling of the polar winter stratosphere, jet

streams weaken and shift poleward. Haigh et al. (2005) analyse the weakening and shift of the subtropical jet using a multiple regression analysis of the NCEP-NCAR reanalysis zonal mean zonal wind velocity. Furthermore, they show with a simple general circulation model that the generic heating of the lower stratosphere tends to weaken the subtropical jets.

In most observational studies, a weakening of the jet is observed over the last decades like Archer & Caldeira (2008) using NCEP and ERA-40 reanalysis sets, Rikus (2015) using ERA-40 data and Molnos et al. (2017) using ERA-Interim data.

However, Pena-Ortiz et al. (2013) found that trends in both strength and position of the jet strongly varies between different reanalysis products.

### 4.2.2    Strength of the storm track activity

In this study, we observe a strengthening of storm track activity under increased global-mean temperature.

Our results are supported by findings from McCabe (2001), who observes a strengthening of the storm track activity with higher global temperature induced by GHG forcing. This is in agreement with Yin (2005) who investigated 15 coupled climate models and showed that storm tracks intensify under global warming. Chang et al. (2012) found that storm tracks in the upper troposphere increase in winter using 23 CMIP5 models (below 300mb they found a slight decreasing).

In addition, the strength of storm track activity depends strongly on the meridional temperature gradient.

This result is intuitive as the prime role of storm tracks within the general circulation is to transport heat poleward, with a stronger temperature contrast leading to enhanced heat transport. It also directly follows from the equation of eddy kinetic energy, which in the first place depends on the meridional temperature gradient (Coumou et al., 2011).

Harvey et al. (2013) observe similar results using CMIP5 data: Larger temperature differences in the equator-to-pole temperature at upper- and lower-tropospheric levels lead to stronger storm activity.

In reanalysis data also a strengthening of the storm tracks can be observed (Schneidereit et al., 2007; Wang et al., 2006), which is supposedly because of the rising global mean temperature. The azonal temperature could not be responsible for a strengthening, since the azonal temperature gradient should be reduced in winter due to global warming. This would lead according to our results to a decreasing strength of the storm track activity.

### 4.2.3    Strength of planetary waves




In our analysis the strengthening of the planetary waves depends on both temperature gradient and global mean temperature. Larger meridional and azonal temperature gradients as well as global mean temperature lead also to stronger winds.

Since azonal wind components emerge due to azonal temperatures, it is expected that higher azonal temperature differences lead to stronger azonal wind components. Stronger temperature gradients cause stronger meridional wind velocities, which

are deflected by the Coriolis force and therefore also the zonal wind velocities are stronger. Those wind velocities are slowed down or accelerated due to topography, land-ocean-contrast and hence also the azonal component of the wind velocity will be stronger. In addition, a higher global mean temperature leads to more available energy in the atmosphere and therefore to a larger azonal wind velocity. The equation for planetary waves is derived from the theory explained above.

Under climate change the global mean temperature increases whereas the meridional temperature is expected to decrease.

Our results suggest that this will have contrary effects on the strength of planetary waves.

Thus, this could explain the results, which Barnes et al. (2013) found by analysing the planetary waves with wave number 1 - 6 as well as wave numbers 1 - 3 using three different data sets. They concluded that there is no significant trend in terms of the strength of the planetary waves and thus arctic amplification does not play a dominant role for changing the undulations of the jet stream.

## 15 5    Conclusion

In this paper, we examined the sensitivity of the width and strength of the storm track activity, jet stream and the Hadley cell as well as the planetary waves to different surface temperature forcing. We used a SDAM, which allows us to do 1000s of individual simulations and thus test the sensitivity of the dynamical fields to different surface temperature changes. This way one can disentangle and separately analyse the effect of global mean temperature, equator-to-pole temperature gradient and

east-west temperature differences. We performed a total of 2601 simulations covering a surface temperature phase space with altered meridional gradients, azonal components and global mean values.

The model's climatology generally reproduces the dynamical fields of ERA-Interim, especially in the Northern Hemisphere, which is the focus of our analysis. We show that the jet streams and storm tracks are more sensitive to changes in the global mean temperature and the meridional temperature gradient than to changes in the azonal temperature gradient. A larger

meridional temperature gradient or a larger global mean value leads to an increased strength of these variables.

This implies that in general the currently observed decreasing meridional temperature gradient associated with Arctic amplification and global warming are likely to have a stronger impact on large-scale atmospheric circulation than more pronounced zonal temperature gradients due to enhanced land-ocean contrast.

This is different for the planetary waves, which depend on all three thermal components. This is expected since azonal wind

components emerge due to azonal temperatures differences. No clear trend in planetary wave strengthening components has been observed, which could be due to opposing effects of rising global mean temperature and decreasing meridional temperature gradient.



In contrast, our results indicate that the width of the Hadley cell generally increases nonlinearly with increasing zonal and meridional temperature gradient. If land masses warm up faster than oceans, the width of the Hadley cell might change in those land areas and that would have an influence on the zonal mean Hadley cell as well.

## Code and data availability

All original data was downloaded from public archives. Code and data are stored in PIK's long term archive, and are made available to interested parties on request.

## Team list

S. Molnos, S. Petri, J. Lehmann, E. Peukert, D. Coumou

## Competing interests

The authors declare that they have no conflict of interest.

## Acknowledgements

We thank ECMWF for making the ERA-Interim available. The work was supported by the German Federal Ministry of Education and Research, grant no. 01LN1304A, (S.M., D.C.). The authors gratefully acknowledge the European Regional Development Fund (ERDF), the German Federal Ministry of Education and Research and the Land Brandenburg for
supporting this project by providing resources on the high performance computer system at the Potsdam Institute for Climate Impact Research.

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

**Figures**

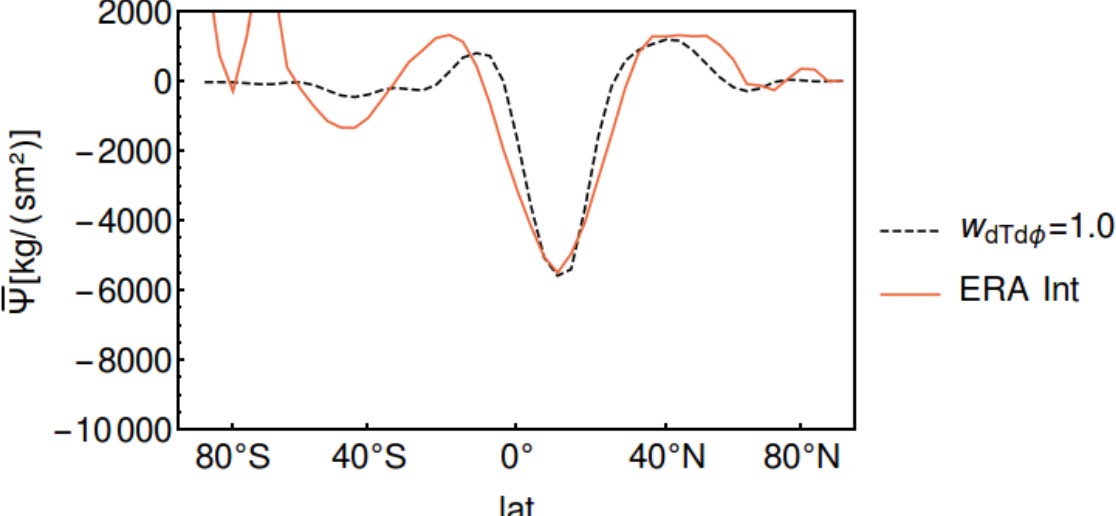

**Fig. 1. Integrated northward mass flux in lower troposphere. Black: Model output for $w_{dTd\phi} = 1$ and $w_{azonal} = 1$. Red: ERA Interim climatological winter data**





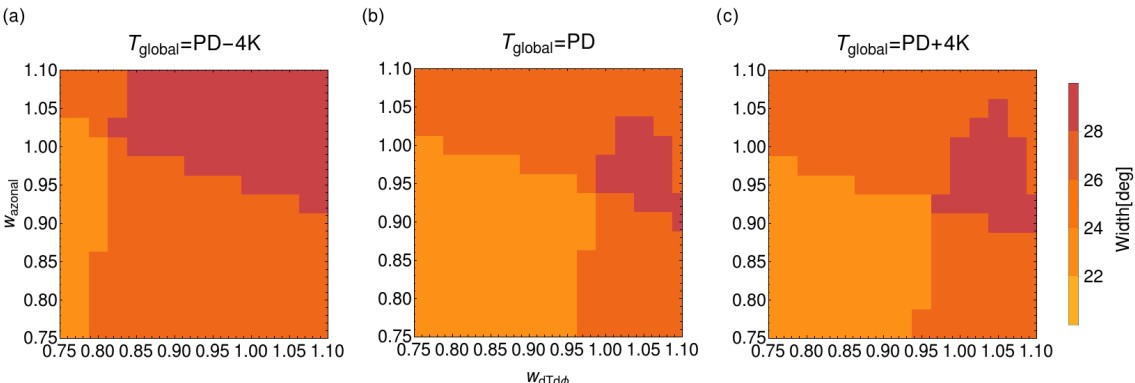

**Fig. 2.** **Width of the Hadley cell depending on $w_{dTd\phi}$ and $w_{azonal}$ for three different global mean temperature values ((a) $T_{global} = \mathrm{PD} - 4\mathrm{K}$ ,(b) $T_{global} = \mathrm{PD}$ and (c) $T_{global} = \mathrm{PD} + 4\mathrm{K}$), whereby PD means present day temperature.**

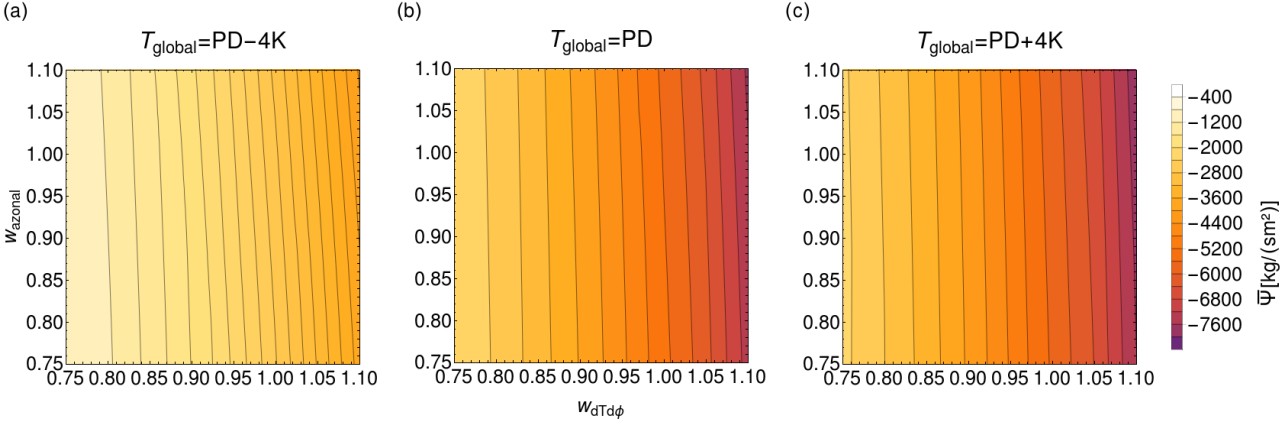

5     **Fig. 3**. **Strength of the Hadley cell depending on $w_{dTd\phi}$ and $w_{azonal}$ for three different global mean temperature values ((a) $T_{global} = \mathrm{PD} - 4\mathrm{K}$ ,(b) $T_{global} = \mathrm{PD}$ and (c) $T_{global} = \mathrm{PD} + 4\mathrm{K}$).**

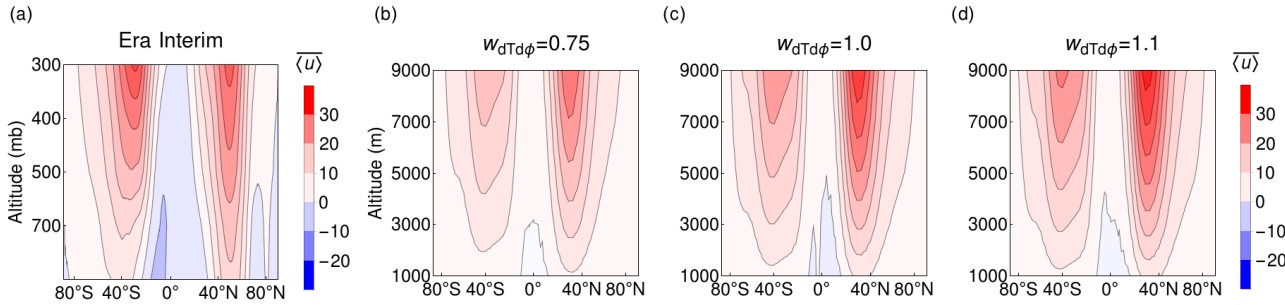

**Fig. 4**. **Zonal mean zonal wind velocity $\overline{\langle u \rangle}$ representing the jet streams. The subpanel (a) shows ERA-Interim data,**
10     **the others results from Aeolus with different $w_{gradient}$. In (b) $w_{dTd\phi} = 0.75$, in (c) $w_{dTd\phi} = 1.0$ and in (d) $w_{dTd\phi} = 1.1$. All other parameters are set to standard values.**





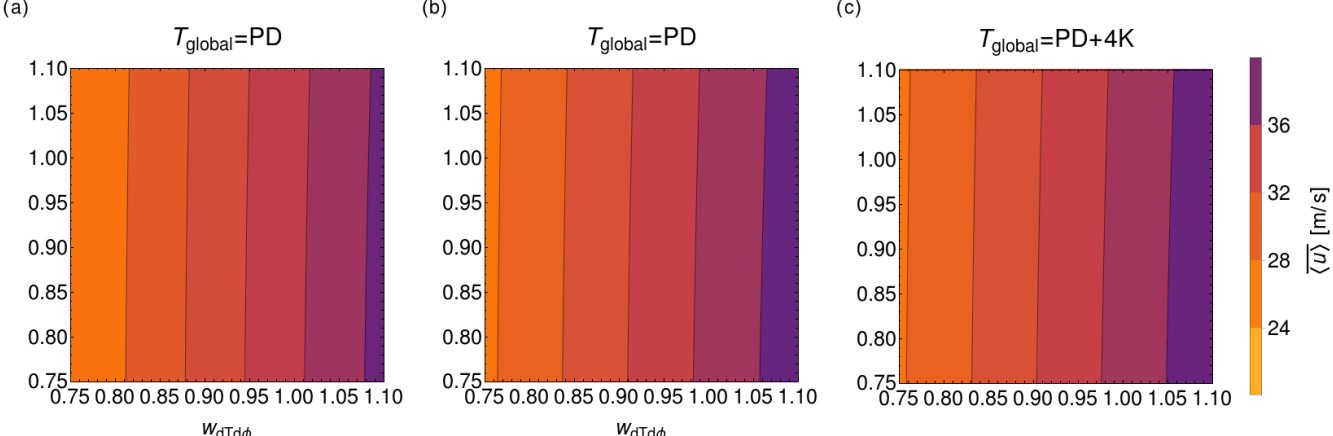

**Fig. 5**. **Jet stream strength defined by the meridional average of the zonal mean zonal wind velocity $\overline{\langle u \rangle}$ between 10°N and 80°N at a height of 9000 m in dependence of $w_{dTd\phi}$ and $w_{azonal}$ for three different values of $T_{Global}$ ((a) $T_{global} =$ PD − 4K ,(b) $T_{global} =$ PD and (c) $T_{global} =$ PD + 4K).**

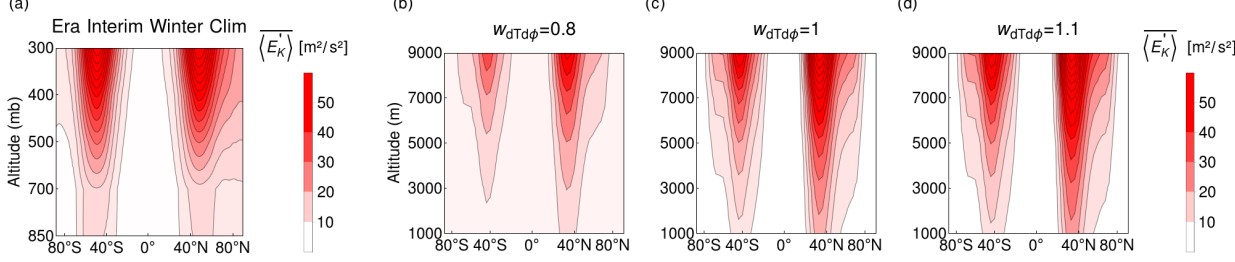

**Fig. 6. Eddy kinetic energy representing storm track activity. Panel (a) shows ERA Interim data, and (b)-(d) from Aeolus with different poleward temperature gradients ($w_{dTd\phi}$). In (b) $w_{dTd\phi} = 0.75$, in (c) $w_{dTd\phi} = 1.0$ and in (d) $w_{dTd\phi} = 1.1$. All other parameters are set to climatology. With larger gradients the storm track acitivty gets stronger.**





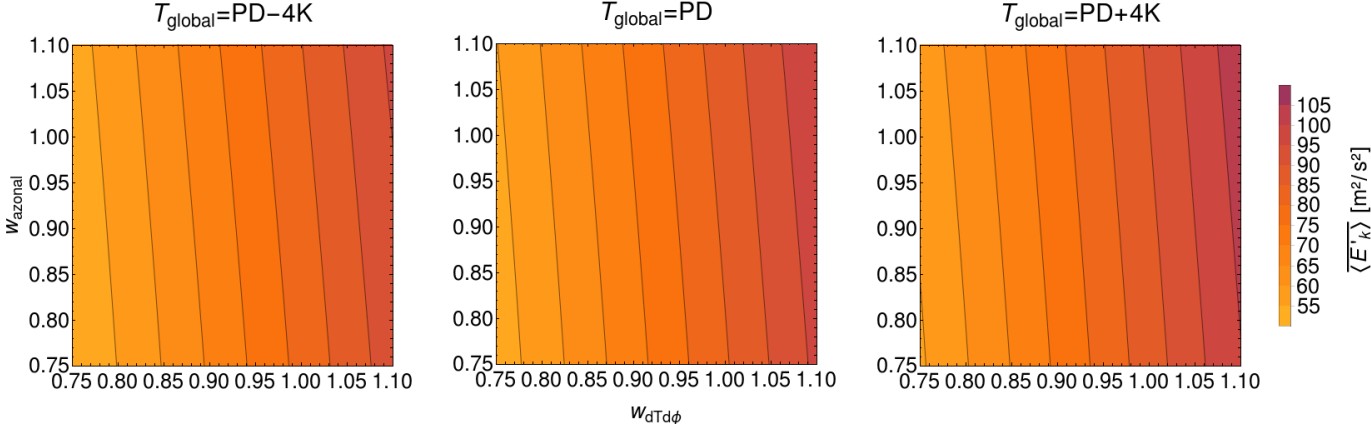

**Fig. 7**. **Strength of storm track activity depending on $w_{dTd\phi}$ and $w_{azonal}$ for three different global mean temperature values ((a) $T_{global} = PD - 4K$ ,(b) $T_{global} = PD$ and (c) $T_{global} = PD + 4K$).**

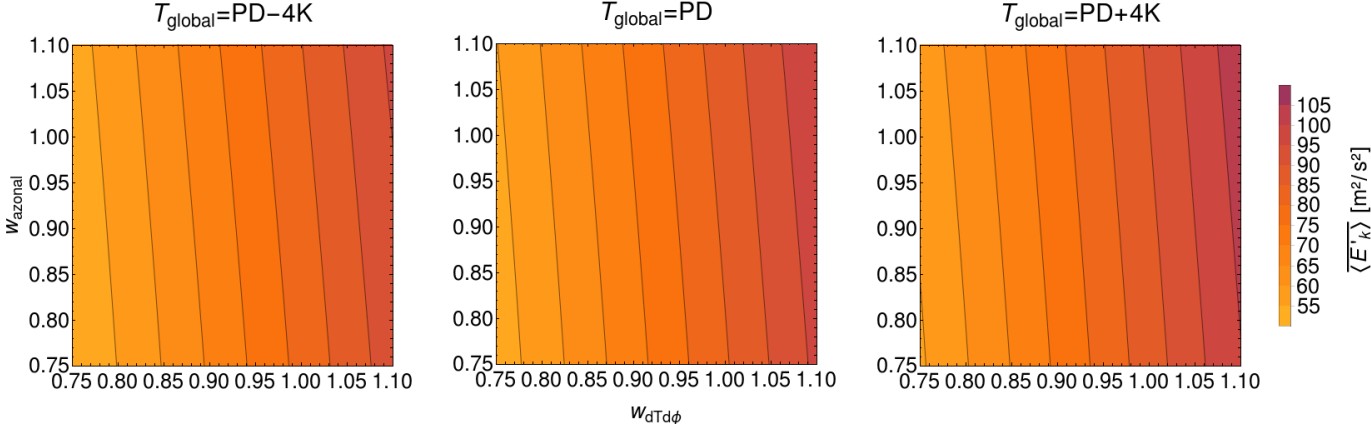

5    **Fig. 8**. **Strength of planetary waves $\langle u^* \rangle$ and $\langle v^* \rangle$ depending on $w_{dTd\phi}$ and $w_{azonal}$ for three different global mean temperature values ((a,d) $T_{global} = PD - 4K$ ,(b,e) $T_{global} = PD$ and (c,f) $T_{global} = PD + 4K$ ).**