# Peer review of "The sensitivity of the large-scale atmosphere circulation to changes in surface temperature gradients in the Northern Hemisphere"

_Earth System Dynamics, 2017_

## Referee Comment (RC1) · Anonymous Referee #1 · 27 Jul 2017

Review of "The sensitivity of the large-scale atmosphere circulation to changes in surface temperature gradients in the Northern Hemisphere" by S. Molnos et al.

The authors use the statistical-dynamical model SDAM to investigate the impact of global, meridional and zonal temperature changes on the large-scale NH atmospheric circulation during boreal winter. I find this a weak and in fact disappointing paper for which I recommend rejection. I base this judgement on concerns regarding the methodology and the quality of results, as well as on editorial concerns.

Methodology:

1. The paper only very briefly describes SDAM. I do have many questions about the

model, however, that the paper misses to address even briefly. Is this a dry model, or does it have some representation of moisture and clouds? How high is the model top? Is there a stratosphere? What about topography? ... All of these are important for the circulation, and it's unclear whether or not these factors are taken into account, and if so how.

2. Temperature perturbations: are the temperature perturbations in the sense of Newtonian background relaxation temperatures, or is this the final temperature. If the latter, it seems the authors are prescribing the u-wind via thermal wind balance, and so prescribe the circulation. At which height are the perturbations applied? This is crucial given the ongoing debate on low-level versus high-level baroclinicity.

3. Circulation metrics: the chosen circulation metrics are unusual, to say the least. This is problematic as it will make comparison to other studies and models difficult, or might even inhibit such comparisons. Two examples: i) the jet stream strength is defined as the meridional average of u between 10N-80N at 9000mb (Fig.5). Why such a choice? Normally it's defined as the maximum zonal wind in the upper troposphere (for the subtropical jet) or the lower troposphere (for the midlatitude eddy-driven jet). ii) the Hadley cell strength is defined as the mass flux between the surface and 500mb. Why, and at which latitude? Normally it is defined as the maximum of the mass stream function. If the mass stream function maximum moves vertically, the the metric of the authors will be unable to take such a shift into account.

Content concerns:

1. There is very little new results in this paper that are of interest beyond the documentation of the SDAM behaviour for this specific setup. Most prominently this is reflected in the abstract, where only three out of 15 sentences are devoted to new results (lines 24-27).

2. The authors claim that they can clearly separate the impact of global, meridional and zonal temperature changes, and that previous studies were unable to do so. But
they entirely neglect the rich literature using dry GCMs that has looked at exactly this question (e.g., papers by Amy Butler, Jian Lu, Janni Yuval, and many more).

Editorial concerns:

The paper reads like a rushed and, to be honest, quite careless write-up. Most figures follow the same layout as if they were all produced with the same script, labels are missing (e.g., y-label in Fig. 5), and the choice of the colormap in the contour plots is poor. There is unnecessary line breaks in the text (e.g., see the introduction). Normally I would not mind, but this slopiness strengthens my feeling that this paper was done in a rush.

I could go on and list more concerns. I think, however, that I have made my point.
* * *

---

## Referee Comment (RC2) · Anonymous Referee #2 · 29 Aug 2017

general comments:

This paper investigates the sensitivity of a statistical-dynamical model to three thermal properties of the atmosphere: meridional and zonal temperature gradients and global temperature. The experiment setup is designed to study the effect of changes in these properties on some tropical and extratropical components of the general circulation.

The paper is well structured and potentially provides some novel insight into the cancelling effects of the observed past and modelled future climate change. It fits well into the theme of ESD. Having said that I have some concerns about the methodology and cannot recommend it for publication at its present state.

[Figure]

specific comments:

- The model temperature forcing is not clear. For example, what is the vertical structure of the forcing? What exactly is $T\_1$ and $T\_new$? I'm not sure what updated temperature means. Is this a restoration temperature field like in the Held-Suarez model?

- Could you include more plots of the zonal asymmetries to support the discussion? I would also put less emphasis on the fact that increasing the meridional temperature gradient increases the jet strength, since this is to be expected from the thermal wind balance.

- How is the separation into planetary and synoptic scale waves done? E.g. are you using the Lanczos filter or just running mean? Also, the planetary wave definition uses departures from the zonal mean, but synoptic eddies are included in this. Discussion of stationary wave changes would be helpful.

- Using a vertically integrated EKE would be more robust, in that it would eliminate the possibility that the changes are due to the EKE maximum moving vertically. some key literature on GCM modelling of the atmospheric response to changes in thermal structure is missing (e.g. work by Tapio Schneider, Paul O'Gorman, Amy Butler, etc.)

technical corrections:

P2 L22: positive trend -> strengthening trend?

P2 L24: reference

P3 L5: the Hadley Cell

P4 L1: chapters -> sections

P5 L4: using 300 hPa to diagnose the jet location: no results for the jet latitude are presented

P5 L21: orange line -> dashed line?

P5 L23: please clarify what you mean by "zero-crossings"

P7 L12: vice versa: it is not clear to me what exactly this is referring to

P7 L12: please clarify what you mean by "wave-shaped structure"

P9 L19: . . .temperature gradient in this study.

P10 L1: gradients

P10 L13: Arctic

P10 L31: . . .observed by previous studies, which. . .

P11 L1: In contrast -> In addition

P11 L2-3: please clarify what you mean by this sentence

Fig 2, 5: Please improve the colour scale

Fig 5: no label on y axis

---

## Short Comment (SC1) · 30 Aug 2017

The paper aims to disentangle the role of global mean, meridional and azonal temperature changes on large scale atmospheric circulation (specifically focusing on jet stream, storm track, planetary wave and Hadley Cell) in the Northern Hemisphere. In order to separate the effect due to each one, the authors carried on simulations with a statistical-dynamical atmosphere model (SDAM) Aeolus 1.0. The authors found that the strength of the Hadley cell, storm tracks and jet streams depends almost linearly on both the global mean temperature and the meridional temperature gradient whereas the zonal temperature gradient has little or no influence.

[Figure]

The paper is potentially interesting, however a lot of fundamental references (see below) are missing in the introduction, and a detailed description of the model and model-set up is needed.

Specific comments:

p2 n.10: The description of the underlying mechanisms controlling the width and the strength of the Hadley Cell is poor. The authors did not mention the static stability and the tropopause height as factors influencing the Hadley Cell (Held, I. M. (2000), The general circulation of the atmosphere. paper presented at 2000 Program in Geophysical Fluid Dynamics, Woods Hole Oceanographic Institution, Woods Hole, Mass., AND Lu et al., 2008 - these references must be included in the main text). Baroclinic instability also plays a role on the strength and extent of the Hadley Cell: please refer to a number of papers by Tapio Schneider.

p3 n. 15: SDAM has been never introduced as acronym before this line (except in the abstract).

p3 n.15 - 20: It is strongly encouraged a detailed description of the model and model-set-up used in this study with a dedicated sub-section in Data and Methods.

p7 n.25 "Our analysis indicates that a higher absolute meridional and azonal temperature gradient leads to a larger Hadley cell width, and we observe only a very weak dependence on the global mean temperature."

In order to improve the discussion of main findings, the author might want also to refer to a new paper in which the strength and the width of the Hadley Cell and its relationship with global mean temperature and meridional temperature gradient has been investigated in a wide range of climate conditions (D'Agostino et al., 2017, http://onlinelibrary.wiley.com/doi/10.1002/2017GL074533/abstract ). Another important study investigating the effect of static stability, tropopause height and meridional temperature gradient in future clime projections on Hadley Cell strength, is "A mechanism

for future changes in Hadley circulation strength in CMIP5 climate change simulations" by Seo et al., 2014 (http://onlinelibrary.wiley.com/doi/10.1002/2014GL060868/full ).

---

## Author Comment (AC1) · 28 Sep 2017

We thank the reviewer for the time she/he took and for the comments provided, which will help us to improve the manuscript. A pointwise reply to the reviewer's comment is given below.

**Methodology**

1.) *The paper only very briefly describes SDAM. I do have many questions about the model, however, that the paper misses to address even briefly. Is this a dry model, or does it have some representation of moisture and clouds? How high is the model top? Is there a stratosphere? What about topography? All of these are important for the circulation, and it's unclear whether or not these factors are taken into account, and if so how?*

We agree with the reviewer that the original manuscript was lacking some details in this respect. However, this can (and we will in a resubmission) be easily improved.
Our model belongs to the class of statistical-dynamical atmosphere models (SDAM) and is called Aeolus 1.0. An outline of Aeolus 1.0 is given in the last paragraph of the introduction with references to earlier papers, which describe the model in more detail (in particular cloud parameterizations and synoptic parameterizations). We appreciate the questions by the reviewer and will add a separate model description section to the manuscript to enhance self-containedness.

Aeolus 1.0 is a wet model. Clouds are represented in the cloud module as written in Molnos et al. (2016) and presented in Eliseev et al. (2013). The dynamical core is coupled with a convective plus 3-layer stratiform cloud scheme which includes low-level, mid-level and upper-level stratiform clouds developed by Eliseev et al. (2013).
However, in this particular experiment, the surface humidity is prescribed to decouple the dynamics from changes in latent heating and associated temperature changes. This way, the dynamical core equilibrates to the prescribed surface temperature patterns without any additional complicating factors.

The equation for humidity is a prognostic equation and described in Petoukhov et al. (2000).

The model has 5 vertical levels in the troposphere with the model top at 10000m altitude. Aeolus 1.0 has a "dummy" stratosphere (i.e. its physics and dynamics are not resolved) to have a boundary condition at the top of the troposphere. In this experiment we excluded topographic influences and it is an atmosphere-only setup using prescribed sea level temperatures. For more information, we refer the reader to Molnos et al. (2016).

2.) *Temperature perturbations: are the temperature perturbations in the sense of Newtonian background relaxation temperatures, or is this the final temperature. If the latter, it seems the authors are prescribing the u-wind via thermal wind balance, and so prescribe the circulation. At which height are the perturbations applied? This is crucial given the ongoing debate on low-level versus high-level baroclinicity*

The temperature perturbation is the final temperature. The temperature perturbations are applied at sea level and propagate to the upper levels based on the lapse rate equation (as schematically shown in Fig. 1). We will add Figure 1 to the manuscript to clarify the process.

Climatological $T_{DJF}$

$T_{DJF} + T_{global}$

$T_{EQ}(\lambda) + w_\phi * (T_{DJF}(\phi, \lambda) - T_{EQ}(\lambda))$

$\overline{T_{DJF}} + w_{azonal} * T^*_{DJF}$

**Figure 1 Schematic plot of the temperature perturbations**

3.) *Circulation metrics: the chosen circulation metrics are unusual, to say the least. This is problematic as it will make comparison to other studies and models difficult, or might even inhibit such comparisons. Two examples: i) the jet stream strength is defined as the meridional average of u between 10N-80N at 9000mb (Fig.5). Why such a choice? Normally it's defined as the maximum zonal wind in the upper troposphere (for the subtropical jet) or the lower troposphere (for the midlatitude eddy-driven jet). ii) the Hadley cell strength is defined as the mass flux between the surface and 500mb. Why,*

*and at which latitude? Normally it is defined as the maximum of the mass stream function. If the mass stream function maximum moves vertically, the the metric of the authors will be unable to take such a shift into account.*

Unfortunately, there has been some confusion on the chosen metrics, and we will improve this throughout the manuscript.

In the following, we explain what metrics we choose and why we choose them:

i.) We calculated the subtropical jet stream as the *maximum* between 10°N-80°N at 9000m altitude. That was phrased incorrectly in Figure 5 and will be corrected. We decided to use 9000m, which corresponds to the 300mb pressure level, since the subtropical jet stream should be located there. This is thus quantitatively in agreement with the reviewer's suggestions.

ii.) The strength of the Hadley cell is defined as the absolute maximum zonal mean integrated mass flux over several latitudes. We believe this is the most appropriate measure and point out that also in the literature there is no consensus on the best metric to use (D'Agostino and Lionello, 2016; Trenberth et al., 2000).

**Content concerns:**

1. *There is very little new results in this paper that are of interest beyond the documentation of the SDAM behaviour for this specific setup. Most prominently this is reflected in the abstract, where only three out of 15 sentences are devoted to new results (lines 24-27)*

The new findings in our manuscript are the following:

- To our knowledge we are the first who systematically sample the global mean temperature and the meridional and azonal temperature gradient in order to receive 3D plots for different investigated variables depending on all three temperature components. Most other studies analyzed the combined effect, e.g. under greenhouse gas scenarios, making it difficult to disentangle cause-effect relationships.
- Although different authors already addressed the question of separating the impact of different precursors, to our knowledge no author used our approach to directly change the global mean temperature, meridional temperature gradient and the azonal temperature gradient as final temperature. Our approach is shown in Fig. 1.
  - Most of the analyzed variables have only a very weak dependence on the azonal component except planetary waves and the width of the

Hadley cell. For that reason it could be that the Hadley cell can widen even though the meridional gradient is the same and only the global temperature changes (Fig. 2).

- o Jet stream, storm tracks & Hadley strength in dependence on the meridional gradient show similar results as found in the literature and are explained in the discussion section, which confirms already existing results. However, in addition, we can find a global temperature dependency in our model.
- o Planetary waves depend on all three temperature components and those results can give one explanation why no significant changes in observational analyses were found by Barnes et al. (2013), since increased global mean temperature and decreased meridional temperature have contrary effects on the strength of planetary waves.

The results in the abstract are only explained briefly. In the revised manuscript we will point out the novelty of our research.

2. *The authors claim that they can clearly separate the impact of global, meridional and zonal temperature changes, and that previous studies were unable to do so. But they entirely neglect the rich literature using dry GCMs that has looked at exactly this question (e.g., papers by Amy Butler, Jian Lu, Janni Yuval, and many more)*

We agree with the reviewer and do not want to claim to be the first to do so. Other authors changed either the CO2 concentration or added a heat source. However, for example using a heat source in the upper troposphere alters not only the global mean temperature but also the azonal and meridional temperature gradient.

Since all below mentioned papers examine either different variables or attributes of the variables (Brewer-Dobson-Circulation, shift of jet streams and shift of storm tracks) or different warming effects (changes in the vertical structure of baroclinicity), we cannot directly compare their results with our results, and cannot include those in the discussion, but we will add the following text in the introduction:

Different studies examined the influences of different temperature sources on the mid-latitude circulation. Butler et al. (2010) addresses the idea to separate the temperature effects by using different heating sources ((1) enhanced warming in the troposphere, (2) enhanced cooling in the stratosphere, (3) enhanced warming at the

surface over the polar region). With their approach Butler et al. can attribute which forcing has the most important influence on the shift of jet streams, storm tracks etc. In a study from 2011 Butler et al. presented an alternative perspective on the response of the mid-latitude tropospheric circulation to zonal-mean tropical heating. The projection of the heating onto the isentropic surfaces at extratropical latitudes drive the poleward shift in wave generation at lower levels. In addition, the poleward shift in the heat fluxes within the troposphere and the diffusive nature of eddy fluxes of the polar vortex lead to a poleward shift in wave breaking near the tropospause.

In addition, Yuval and Kaspi (2016) investigated changes in the vertical structure of baroclinicity to the magnitude of eddy kinetic energy and eddy fluxes using an idealized global circulation model. This is especially interesting, since new studies indicate that under increasing $CO_2$ concentrations the lower-tropospheric temperature gradient will decrease whereas the upper tropospheric temperature gradient will increase with counteracting effects on eddy activity. The results demonstrate that eddy activity is more sensitive to temperature gradient changes in the upper troposphere.

Moreover, Shaw and Voigt (2015) examined the radiative changes of clouds and water vapour for two different aquaplanet climate models and found that they are important to the regional response of precipitation and atmosphere circulation. They concluded that uncertainty in circulation is linked to uncertainty in the behaviour of clouds and water vapour.

We will add the following text in the discussion:

Lu et al. (2007) found a robust weakening and a poleward expansion of the Hadley circulation in response to increased GHG forcing in simulations of the 21st century climate taken from the A2 scenario of the IPCC AR4 project (Lu et al., 2007). Lu et al. (2008) analyzed the change in the zonal mean atmospheric circulation under global warming in comparison with the response to El Niño forcing, by examining the CMIP5 model simulations. They used again the A2 scenario to simulate global warming. Under global warming due to higher $CO_2$ concentrations the Hadley cell weakens and expands northwards together with a poleward shift of the jet stream.

Based on our results, we can assume that "El Niño–like" enhanced warming leads to a stronger meridional temperature gradient (and a higher global mean temperature) resulting in a stronger Hadley cell, whereas the $CO_2$ concentration leads to a weaker meridional temperature gradient (and a higher global mean temperature) and as a consequence the Hadley cell weakens. This can also explain the widening of the

Hadley cell, which we observe in our experiments as well: A stronger meridional temperature gradient can lead to a smaller width of the Hadley cell and vice versa.

**Editorial concerns:**

1. *The paper reads like a rushed and, to be honest, quite careless write-up. Most figures follow the same layout as if they were all produced with the same script, labels are missing (e.g., y-label in Fig. 5), and the choice of the colormap in the contour plots is poor. There is unnecessary line breaks in the text (e.g., see the introduction). Normally I would not mind, but this slopiness strengthens my feeling that this paper was done in a rush.*

The label mentioned in Fig.5 (a) was added incorrectly. The y-label should occur only in the center figure.

We will improve the colorbar and plot 3D plots to visualize the experiments (Fig 2 - 6).

[Figure]

**Figure 2 Width of the Hadley cell in dependence of $w_{T_\phi}$ and $w_{azonal}$ and $\Delta T_{G,PD}$, whereby $\Delta T_{G,PD}$ is the difference between the present day temperature and the changed global mean temperature.**

[Figure]

**Figure 3** Strength of the Hadley cell in dependence of $w_{T_\phi}$ and $w_{azonal}$ and $\Delta T_{G,PD}$, whereby $\Delta T_{G,PD}$ is the difference between the present day temperature and the changed global mean temperature. The arrow points in the direction of the strongest gradient.

[Figure]

**Figure 4** Jet stream strength defined by the meridional average of the zonal mean zonal wind velocity ($\overline{\langle u \rangle}$) between 10°N and 80°N at a height of 9000 m in dependence of $w_{T_\phi}$ and $w_{azonal}$ and $\Delta T_{G,PD}$, whereby $\Delta T_{G,PD}$ is the difference between the present day temperature

and the changed global mean temperature. The arrow points in the direction of the strongest gradient.

[Figure]

Figure 5 Strength of storm track activity in dependence of $w_{T_\phi}$ and $w_{azonal}$ and $\Delta T_{G,PD}$, whereby $\Delta T_{G,PD}$ is the difference between the present day temperature and the changed global mean temperature. The arrow points in the direction of the strongest gradient.

[Figure]

Figure 6 Strength of planetary waves $\langle u^* \rangle$ and $\langle v^* \rangle$ in dependence of $w_{T_\phi}$ and $w_{azonal}$ and $\Delta T_{G,PD}$, whereby $\Delta T_{G,PD}$ is the difference between the present day temperature and the changed global mean temperature. The arrow points in the direction of the strongest gradient.

**References**

Butler, A., Thompson, W. J. and Heikes, R.: The Steady-State Atmospheric Circulation Response to Climate Change – like Thermal Forcings in a Simple General Circulation Model, J. Clim., 3474–3496, doi:10.1175/2010JCLI3228.1, 2010.

Butler, A. H., Thompson, D. W. J. and Birner, T.: Isentropic Slopes , Downgradient Eddy Fluxes , and the Extratropical Atmospheric Circulation Response to Tropical Tropospheric Heating, J. Atmos. Sci., 68(2007), 2292–2305, doi:10.1175/JAS-D-10-05025.1, 2011.

Coumou, D., Petoukhov, V. and Eliseev, A. V.: Three-dimensional parameterizations of the synoptic scale kinetic energy and momentum flux in the Earth's atmosphere, Nonlinear Process. Geophys., 18(6), 807–827, doi:10.5194/npg-18-807-2011, 2011.

D'Agostino, R. and Lionello, P.: Evidence of global warming impact on the evolution of the Hadley Circulation in ECMWF centennial reanalyses, Clim. Dyn., 48(9), 1–14, doi:10.1007/s00382-016-3250-0, 2016.

Eliseev, A. V., Coumou, D., Chernokulsky, A. V., Petoukhov, V. and Petri, S.: Scheme for calculation of multi-layer cloudiness and precipitation for climate models of intermediate complexity, Geosci. Model Dev., 6(5), 1745–1765, doi:10.5194/gmd-6-1745-2013, 2013.

Lu, J., Vecchi, G. A. and Reichler, T.: Expansion of the Hadley cell under global warming, Geophys. Res. Lett., 34(October 2006), 2–6, doi:10.1029/2006GL028443, 2007.

Lu, J., Chen, G. and Frierson, D. M. W.: Response of the Zonal Mean Atmospheric Circulation to El Niño versus Global warming, J. Clim., doi:10.1175/2008JCLI2200.1, 2008.

Molnos, S., Eliseev, A. V., Petri, S., Flechsig, M., Caesar, L. and Petoukhov, V.: The Dynamical Core of the Aeolus Statistical - Dynamical Atmosphere Model : Validation and Parameter Optimization, Geosci. Model Dev. Discuss., doi:10.5194/gmd-2016-263, 2016.

Trenberth, K. E., Stepaniak, D. P. and Caron, J. M.: The global monsoon as seen through the divergent atmospheric circulation, J. Clim., 13(22), 3969–3993, doi:10.1175/1520-0442(2000)013<3969:TGMAST>2.0.CO;2, 2000.

Yuval, J. and Kaspi, Y.: Eddy Activity Sensitivity to Changes in the Vertical Structure of Baroclinicity, J. Atmos. Sci., 73, 1709–1726, doi:10.1175/JAS-D-15-0128.1, 2016.

---

## Author Comment (AC2) · 28 Sep 2017

We thank the reviewer for the time she/he took and for the comments provided, which will help us to improve the manuscript. A pointwise reply to the reviewer's comment is given below.

**Specific comments**

1.) *The model temperature forcing is not clear. For example, what is the vertical structure of the forcing? What exactly is T_1 and T_new? I'm not sure what updated temperature means. Is this a restoration temperature field like in the Held-Suarez model??*

The temperature $T_{New}$ is the final temperature based on all three temperature components and is used as model input to which the dynamical core equilibrates. In temperature $T_1$ only the global mean temperature and the meridional temperature gradient is altered/ updated. The azonal temperature gradient is not yet included (as its influence on global-mean temperature is zero as opposing azonal temperature anomalies cancel each other out).
We will rewrite it in: $T_1(\phi, \lambda)$ is the altered temperature with changed global mean temperature and meridional temperature.

2.) *Could you include more plots of the zonal asymmetries to support the discussion? I would also put less emphasis on the fact that increasing the meridional temperature gradient increases the jet strength, since this is to be expected from the thermal wind balance.*

We agree with the reviewer and will include 3D plots with all different azonal components used in the experiments in order to support the discussion:

[Figure]

**Figure 1** Width of the Hadley cell in dependence of $w_{T_\phi}$ and $w_{azonal}$ and $\Delta T_{G,PD}$, whereby $\Delta T_{G,PD}$ is the difference between the present day temperature and the changed global mean temperature.

[Figure]

**Figure 2** Strength of the Hadley cell in dependence of $w_{T_\phi}$ and $w_{azonal}$ and $\Delta T_{G,PD}$, whereby $\Delta T_{G,PD}$ is the difference between the present day temperature and the changed global mean temperature. The arrow points in the direction of the strongest gradient.

[Figure]

**Figure 3 Jet stream strength defined by the meridional average of the zonal mean zonal wind velocity ($\langle u \rangle$ )¯ between 10°N and 80°N at a height of 9000 m in dependence of $w_{T_\phi}$ and $w_{azonal}$ and $\Delta T_{G,PD}$, whereby $\Delta T_{G,PD}$ is the difference between the present day temperature and the changed global mean temperature. The arrow points in the direction of the strongest gradient.**

[Figure]

**Figure 4 Strength of storm track activity in dependence of $w_{T_\phi}$ and $w_{azonal}$ and $\Delta T_{G,PD}$, whereby $\Delta T_{G,PD}$ is the difference between the present day temperature and the changed global mean temperature. The arrow points in the direction of the strongest gradient.**

[Figure]

**Figure 5 Strength of planetary waves $\langle u^* \rangle$ and $\langle v^* \rangle$ in dependence of $w_{T_\phi}$ and $w_{azonal}$ and $\Delta T_{G,PD}$, whereby $\Delta T_{G,PD}$ is the difference between the present day temperature and the changed global mean temperature. The arrow points in the direction of the strongest gradient.**

We will put less emphasis on the fact that increasing meridional gradient increases the jet strength.

3.) *How is the separation into planetary and synoptic scale waves done? E.g. are you using the Lanczos filter or just running mean? Also, the planetary wave definition uses departures from the zonal mean, but synoptic eddies are included in this. Discussion of stationary wave changes would be helpful.*

In the atmosphere model Aeolus 1.0, the synoptic waves are parameterized in terms of the large scale wind field which is the basic idea of a statistical-dynamical method. A full description can be found in Coumou et al. (2011). The equations for planetary waves are derived in Molnos et al. (2016). We will add this info in the revised manuscript.

For comparison, we calculated planetary and synoptic waves from zonal and meridional wind fields using ERA-Interim data. The synoptic waves were calculated by using a 2.5-6 bandpass filter. The planetary waves are the azonal deviations from the monthly-mean zonal mean wind field.

4.) *Using a vertically integrated EKE would be more robust, in that it would eliminate the possibility that the changes are due to the EKE maximum moving vertically.*

We agree with the reviewer and used a vertical integrated EKE. The results are qualitatively similar (Fig. 5).

5.) *Some key literature on GCM modelling of the atmospheric response to changes in thermal structure is missing (e.g. work by Tapio Schneider, Paul O'Gorman, Amy Butler, etc.)*

We agree with the reviewer and will include additional literature:

In the introduction summary we will include:
Moreover, recent studies using dry and most idealized general circulation models suggest that the strength and extent of the Hadley cell depends on static stability, meridional temperature gradient and the tropopause level (Schneider and Walker, 2008; O'Gorman, 2011; Levine and Schneider, 2015).

In the storm tracks section we will include:
O'Gorman and Schneider (2008) examined the response of storm tracks to different climate conditions simulating an aquaplanet model and by changing the longwave

optical thickness in the radiation scheme of the GCM (representing variations in greenhouse gas concentrations). They found that eddy kinetic energy has a maximum for a climate with the global-mean temperature similar to that of present-day-climate. Lower or higher global-mean temperatures lead to significantly smaller values. In addition, they observed that the eddy kinetic energy increases monotonically with the meridional insolation gradient (representing changes in, for example, high-latitude surface albedo).

Similarly, Pfahl et al. (2015) investigated the behavior of extratropical cyclones under strongly varying climate conditions using idealized climate model simulations in an aquaplanet setup. They changed the meridional insolation gradient together with the longwave optical thickness with shortwave parameters held constant. They found that the maximum of eddy kinetic energy is reached at a global mean temperature slightly warmer than present-day climate.

These results are different to our results, where no such peak in EKE can be observed. The different results may be explained by the different techniques applied to simulate higher global mean temperature. In our study, we directly change the temperature, whereas Pfahl et al. change the longwave optical thickness with shortwave parameters held constant, which represents variations in longwave absorbers like carbon dioxide and water vapor. These changes could also change the meridional and azonal temperature gradient leading to different results.

However, we also observe a strong positive dependence between temperature gradient and Eddy kinetic energy.

In the Hadley cell section, we will include:

Seo et al. (2014) investigated possible drivers of the Hadley cell such as the meridional temperature gradient, gross static stability and tropopause height using the Coupled Model Intercomparison Project Phase 5 (CMIP5). Consistent with our results, they found a robust dependence between meridional temperature gradient and the strength of the Hadley cell in winter: A decreased meridional temperature gradient leads to a weakening of the Hadley cell.

In addition, D'Agostino et al. (2017) analyzed and compared the Hadley cell during the last glacial maximum to global warming scenarios (RCP4.5 and RCP8.5) with a focus on dependence on subtropical stability, near-surface meridional potential temperature gradient, and the tropical tropopause level. They concluded that the meridional temperature gradient is a major driver for Hadley cell changes.

However, in both studies the atmospheric composition in terms of anthropogenic aerosols is changed and hence not only the meridional temperature gradient changes but also the global mean temperature and the azonal temperature gradient. This makes it difficult to attribute changes in the Hadley cell to one temperature component only.

**Specific comments**

1.) *P2 L22:positive trend -> strengthening trend?*
Thanks, we will correct this.

2.) *P2 L24: reference Storm tracks play a crucial role in modulating precipitation in the Earth system.*
Thanks, we will add the references (Raible *et al.*, 2007; Hawcroft *et al.*, 2012; Lehmann and Coumou, 2015)

3.) *P3 L5: the Hadley Cell*
Thanks we corrected it.

4.) *P4 L1: chapters -> sections*
Thanks, we corrected it.

5.) *P5 L4: using 300 hPa to diagnose the jet location: no results for the jet latitude are presented*
We removed this sentence.

6.) *P5 L21: orange line -> dashed line?*
Yes dashed line, we will correct this.

7.) *P5 L23: please clarify what you mean by "zero-crossings"*
Zero-crossing refers to the point in the graph where the function f(x) crosses the x=0 line. We will rewrite this part in the revised manuscript.

8.) *P7 L12: vice versa: it is not clear to me what exactly this is referring to*
We want to make the statement that if the azonal temperature gradient is smaller than the meridional temperature gradient, the strength of the planetary waves increases faster with higher azonal temperature gradient. We will rewrite this part in the revised manuscript.

9.) *P7 L12: please clarify what you mean by "wave-shaped structure"*
Wave-shaped structure means that the mean strength of the planetary waves varies in a curved way depending on meridional temperature gradient and azonal temperature gradient. We will rewrite this part.

10.) *P9 L19: : : :temperature gradient in this study.*
Thanks, we will add that.

11.) *P10 L1: gradients*

Thanks, we will correct that.

*12.)        P10 L13: Arctic*
        Thanks, we will correct that.

*13.)        P10 L31: : : :observed by previous studies, which: : :*
        Thanks we will add this.

*14.)        P11 L1: In contrast -> In addition*
        We will change this.

15.)        P11 L2-3: please clarify what you mean by this sentence
        This sentence means that the Hadley cell might have a longitudinal dependence (due to land masses and ocean), which can explain the dependence of the Hadley cell width on the azonal component. We will rewrite this part.

*16.)        Fig 2, 5: Please improve the colour scale*
        We will improve the colorbar (as shown in the response of reviewer 1)

*17.)        Fig 5: no label on y axis*
        Fig 5 is going to be replaced by the 3-D version that is shown in
        the response to reviewer #1.

References:

Coumou, D., Petoukhov, V. and Eliseev, A. V. (2011) 'Three-dimensional parameterizations of the synoptic scale kinetic energy and momentum flux in the Earth's atmosphere', *Nonlinear Processes in Geophysics*, 18(6), pp. 807–827. doi: 10.5194/npg-18-807-2011.

D'Agostino, R., Lionello, P., Adam, O. and Schneider, T. (2017) 'Factors controlling Hadley circulation changes from the Last Glacial Maximum to the end of the 21st century', *Geophysical Research Letters*, 44, pp. 1–7. doi: 10.1002/2017GL074533.1.

Hawcroft, M. K., Shaffrey, L. C., Hodges, K. I. and Dacre, H. F. (2012) 'How much Northern Hemisphere precipitation is associated with extratropical cyclones?', *Geophysical Research Letters*, 39(24), pp. 1–7. doi: 10.1029/2012GL053866.

Lehmann, J. and Coumou, D. (2015) 'The influence of mid-latitude storm tracks on hot, cold, dry and wet extremes.', *Scientific reports*, 5, p. 17491. doi: 10.1038/srep17491.

Levine, X. J. and Schneider, T. (2015) 'Baroclinic Eddies and the Extent of the Hadley Circulation : An Idealized GCM Study', *Journal of Atmospheric Sciences*, 72, pp. 2744–2761. doi: 10.1175/JAS-D-14-0152.1.

Molnos, S., Eliseev, A. V., Petri, S., Flechsig, M., Caesar, L. and Petoukhov, V. (2016) 'The

Dynamical Core of the Aeolus Statistical - Dynamical Atmosphere Model : Validation and Parameter Optimization', *Geoscience Model Developement Discussion*. doi: 10.5194/gmd-2016-263.

O'Gorman, P. A. (2011) 'The Effective Static Stability Experienced by Eddies in a Moist Atmosphere', *Journal of Atmospheric Sciences*, 68, pp. 75–90. doi: 10.1175/2010JAS3537.1.

O'Gorman, P. A. and Schneider, T. (2008) 'Energy of Midlatitude Transient Eddies in Idealized Simulations of Changed Climates', *Journal of Climate*, 21, pp. 5797–5806. doi: 10.1175/2008JCLI2099.1.

Pfahl, S., O'Gorman, P. A. and Singh, M. S. (2015) 'Extratropical Cyclones in Idealized Simulations of Changed Climates', *Journal of Climate*, 28, pp. 9373–9392. doi: 10.1175/JCLI-D-14-00816.1.

Raible, C. C., Yoshimori, M., Stocker, T. F. and Casty, C. (2007) 'Extreme midlatitude cyclones and their implications for precipitation and wind speed extremes in simulations of the Maunder Minimum versus present day conditions', *Climate Dynamics*, 28(4), pp. 409–423. doi: 10.1007/s00382-006-0188-7.

Schneider, T. and Walker, C. C. (2008) 'Scaling Laws and Regime Transitions of Macroturbulence in Dry Atmospheres', *Journal of Atmospheric Sciences*, 65, pp. 2153–2173. doi: 10.1175/2007JAS2616.1.

Seo, K., Frierson, D. M. W. and Son, J. (2014) 'A mechanism for future changes in Hadley circulation strength in CMIP5 climate change simulations', pp. 5251–5258. doi: 10.1002/2014GL060868.Received.

Shaw, T. A. and Voigt, A. (2015) 'Tug of war on summertime circulation between radiative forcing and sea surface warming', *Nature Geoscience*, 8(7), pp. 560–566. doi: 10.1038/ngeo2449.

---

## Author Comment (AC3) · 28 Sep 2017

Thanks a lot for your short comment, which will certainly improve the manuscript. We will include all suggested literature as written in the response of reviewer 1 and reviewer 2. We will also include a description of the model and introduce the acronym SDAM before p. 3 l.15.